# SEQ-VCR: PREVENTING COLLAPSE IN INTERMEDIATE TRANSFORMER REPRESENTATIONS FOR ENHANCED REASONING

**Md Rifat Arefin**[1,2,3*] **Gopeshh Subbaraj**[1,2]**, Nicolas Gontier**[3]**, Yann LeCun**[5,6]**,
Irina Rish**[1,2]**, Ravid Shwartz-Ziv**[6]**, Christopher Pal**[3,4]

[1]Université de Montréal, [2]Mila, [3]ServiceNow, [4]Polytechnique Montreal, [5]Meta FAIR,
[6]New York University

## ABSTRACT

Decoder-only Transformers often struggle with complex reasoning tasks, particularly arithmetic reasoning requiring multiple sequential operations. In this work, we identify *representation collapse* in the model's intermediate layers as a key factor limiting their reasoning capabilities. To address this, we propose **Sequential Variance-Covariance Regularization (Seq-VCR)**[1], which enhances the entropy of intermediate representations and prevents collapse. Combined with dummy pause tokens as substitutes for chain-of-thought (CoT) tokens, our method significantly improves performance in arithmetic reasoning problems. In the challenging $5 \times 5$ integer multiplication task, our approach achieves $99.5\%$ exact match accuracy, outperforming models of the same size (which yield $0\%$ accuracy) and GPT-4 with five-shot CoT prompting ($44\%$). We also demonstrate superior results on arithmetic expression and longest increasing subsequence (LIS) datasets. Our findings highlight the importance of preventing intermediate layer representation collapse to enhance the reasoning capabilities of Transformers and show that Seq-VCR offers an effective solution without requiring explicit CoT supervision.

# 1 INTRODUCTION

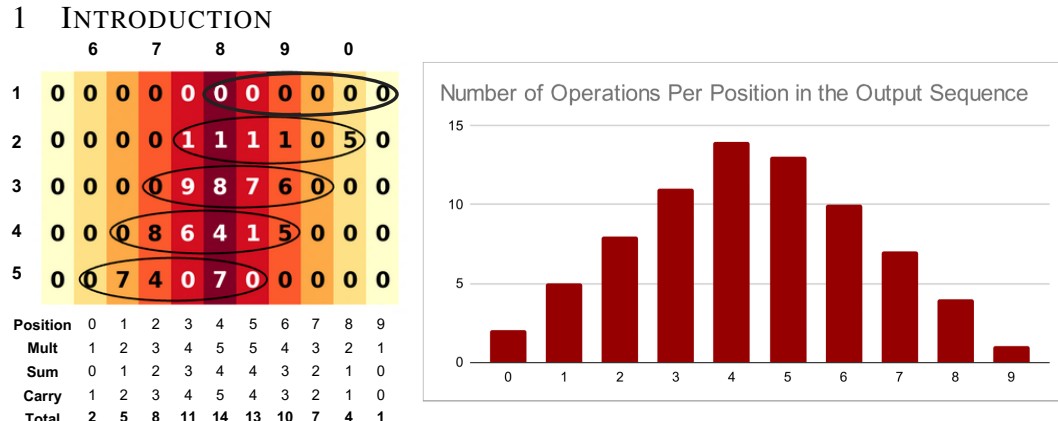

Figure 1: Position-wise number of operations needed for 5x5 digits integer multiplication task. Middle tokens in the output sequence need more operations than the peripheral ones, making their prediction much harder (as shown in Figure 6). Example of 12345 x 67890 is shown here.

Large Language Models (LLMs) based on Transformer architectures have achieved remarkable success across a wide range of tasks, positioning them as foundational models in artificial intelligence (Bommasani et al., 2021). Despite their impressive capabilities, LLMs often struggle with tasks

---

*rifat.arefin@mila.quebec, Work was completed during internship at ServiceNow.

[1]https://github.com/rarefin/seq_vcr

requiring complex reasoning, particularly arithmetic reasoning that necessitates multiple sequential operations (Bubeck et al., 2023). These challenges are attributed to the models' limitations in handling tasks that involve deep cognitive abilities and multi-step reasoning processes.

One of the key obstacles is the *representation collapse* in the intermediate layers of Transformer models. Representation collapse occurs when internal representation diversity diminishes, leading to less informative features and hindering the model's ability to solve complex tasks (Jing et al., 2021). For arithmetic reasoning, transformers struggle with successive carryovers and storing intermediate results Qiu et al. (2024), which are essential for solving complex sub-tasks, requiring more computations (Figure 1). We hypothesize that representation collapse prevents the model from effectively performing sub-tasks by calculating successive carryovers and storing intermediate results, which are essential for accurate prediction.

To address this limitation, we introduce **Sequential Variance-Covariance Regularization (Seq-VCR)**, a regularization technique designed to enhance the entropy of intermediate representations and prevent representation collapse. By increasing the diversity of representations within the model's layers, Seq-VCR enables the Transformer to maintain richer and more informative features throughout the computation process.

Furthermore, we incorporate dummy pause tokens as substitutes for chain-of-thought (CoT) tokens. While CoT prompting has been shown to improve reasoning by breaking down tasks into intermediate steps (Wei et al., 2022), it often requires explicit supervision and can be computationally expensive. Our approach leverages pause tokens to simulate the effect of CoT without the need for explicit intermediate reasoning steps.

We validate our method on challenging arithmetic reasoning tasks. Notably, on the $5 \times 5$ integer multiplication task, our approach achieves $99.5\%$ exact match accuracy, surpassing models of the same size (which yield $0\%$ accuracy) and GPT-4 with five-shot CoT prompting ($44\%$). We also demonstrate significant improvements on arithmetic expression and longest increasing subsequence (LIS) datasets.

Our contributions are summarized as follows:

- We identify representation collapse in intermediate layers as a key limitation affecting the reasoning capabilities of Transformer models.

- We propose Seq-VCR, a regularization technique that enhances the diversity of intermediate representations and prevents collapse.

- We demonstrate that combining Seq-VCR with pause tokens enables models to solve complex arithmetic reasoning tasks without explicit CoT supervision.

- We provide extensive experimental results showing significant improvements over baseline models and state-of-the-art LLMs like GPT-4.

The paper is structured as follows: **Section 2** reviews reasoning in large language models, representation collapse, and regularization. **Section 3** presents our method, Sequential Variance-Covariance Regularization (Seq-VCR), including its formulation and motivation. **Section 4** describes the experimental setup, datasets, models, and shows Seq-VCR's effectiveness in arithmetic reasoning and its impact on representations. **Section 5** summarizes key findings and discusses future directions.

## 2 LITERATURE REVIEW

**Transformer Reasoning** Research in large language models (LLMs) has advanced significantly. Models like BERT (Devlin et al., 2019) and GPT (Radford & Narasimhan, 2018) demonstrated improvements in natural language understanding. However, challenges remain in tasks that require deep cognitive abilities. Integrating external knowledge has shown promise in improving reasoning capabilities (Bosselut et al., 2019). Recent advances include techniques such as chain-of-thought (CoT) prompting, which allow models such as GPT-3 (Brown et al., 2020) to perform complex reasoning by breaking tasks into intermediate steps (Wei et al., 2022) with human supervision, called process supervision (Lightman et al., 2023). Wang et al. (2022) shows that even incorrect but coherent intermediate steps can improve reasoning performance. Jin et al. (2024) found increasing the

length of the reasoning steps in the prompts, even without adding new information to the prompt, improves the reasoning abilities of LLMs.

Wang et al. (2024), investigated the addition of dummy tokens such as periods or hash symbols to inputs at inference time. They found that this simple modification could improve performance in arithmetic reasoning tasks. Some works show that these dummy tokens, commonly referred to as filler tokens, do not extend transformers' abilities beyond $TC^0$ circuit complexity, but still significantly enhance problem-solving within this class (Merrill & Sabharwal, 2023; Strobl et al., 2023). Building on this idea, Goyal et al. (2023a) introduced a more comprehensive framework called "pause-training." Their approach involves incorporating learnable $<pause>$ tokens during both pre-training and fine-tuning of language models. While there are differences between the pause tokens and filler tokens approach, they share the goal of providing additional computation time for transformers which extends the expressive power of transformers within the $TC^0$ class without changing the fundamental limitations of the model architecture. Chain-of-thought prompting can potentially elevate transformers beyond the $TC^0$ complexity class; However, it may not be necessary for all problems. Furthermore, filler tokens have been demonstrated to enhance the expressivity of transformers (Pfau et al., 2024), even in cases where traditional transformers, lacking chain-of-thought mechanisms, exhibit insufficient expressive power (Sanford et al., 2024).

**Representation Collapse**  Representation collapse in representation learning degrades model performance by making features indistinguishable. It often occurs in unsupervised and self-supervised learning tasks due to the lack of labeled data, leading to trivial solutions and a low-rank feature space. Research by Jing et al. (2021) highlights that representation collapse is due to poor optimization, loss function design, and improper regularization. Recent studies (Grill et al., 2020) suggest the use of contrastive learning, such as SimCLR, to mitigate this by contrasting positive and negative samples. Additionally, Zbontar et al. (2021) introduced Barlow Twins to minimize redundancy between learned representations. A further advancement is VICReg (Bardes et al., 2021; Zhu et al., 2023) which employs variance, invariance, and covariance regularization to prevent collapse, ensuring diverse and non-trivial representations. While these researches focus on images, it is less explored in language models. Recently, Barbero et al. (2024) studied last-layer representations of the next tokens and found representation collapse in pre-trained models.

## 3 METHODOLOGY

### 3.1 PRELIMINARIES

Consider a finite set of token vocabulary denoted by $V = (1 \dots v)$, while $x, y \in V$ constitutes a sequence of these tokens representing the input $\mathbf{x} = [x_1, \dots, x_K]$ and the output $\mathbf{y} = [y_1, \dots, y_T]$, where $K$ and $T$ indicate the respective sequence lengths of the input and output.

We covert the input $\mathbf{x}$ and output sequence $\mathbf{y}$ in an embedded sequence:

$E_{emb} = [\tilde{x}_1, \dots, \tilde{x}_K, \tilde{y}_1, \dots, \tilde{y}_T] \in \mathbb{R}^{T+K \times d}$ and $E_{pos} = [p_1, p_2, \dots] \in \mathbb{R}^{T+K \times d}$ of dimension $d$.

We can get the language model layer-wise as: $f = (f_{cls} \circ f_L \circ f_{L-1} \dots \circ f_0)$

where $f_0 = [\tilde{x}_1 + p_1, \dots, \tilde{x}_K + p_K, \tilde{y}_1 + p_{K+1}, \dots, \tilde{y}_T + p_{K+T}]$ and $f_{cls} \in \mathbb{R}^{|V| \times d}$ is the final linear classification layer. $f_l$ is the self-attention layer with MLP for layer $l$ (Vaswani et al., 2017).

### 3.2 NEXT-TOKEN PREDICTION LOSS

Large language models like *GPT* (Radford et al., 2018) (Generative Pre-trained Transformer) are trained to auto-regressively predict the next token in a sequence. This minimizes the difference between predicted and actual tokens for coherent and contextually appropriate text. A loss function like cross-entropy for next-token prediction is given by:

$$\mathcal{L}_{next}(y, f(x)) = -\sum_{t=1}^{T} y_t^\top \log(f(x, y_{<t})), \tag{1}$$

where $y_t$ is the one-hot encoded true token at time step $t$ and input tokens $x$, modeled by a language model, $f$.

## 3.3 Representation Collapse in Transformers

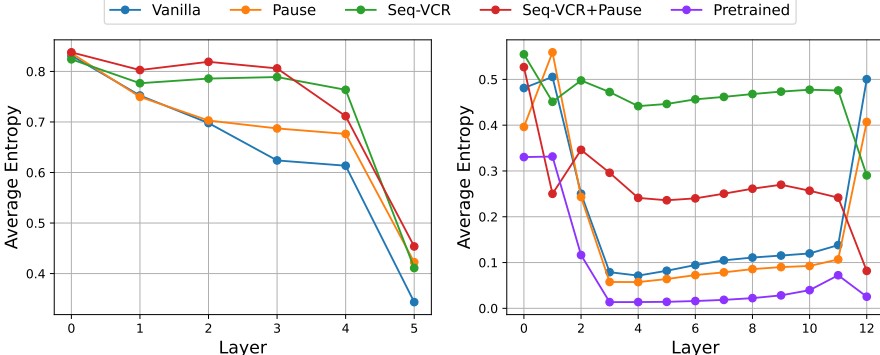

(a) Training minGPT from scratch on *Arithmetic Expression*

(b) Fine-tuning GPT-2 Small on $5 \times 5$ *digit Multiplication*

Figure 2: **Representation collapse** across layers during (a) training (or fine-tuning(b)) for two datasets. The x-axis represents the layer ID, while the y-axis shows the degree of collapse as measured by representation Matrix-Entropy. The results highlight how intermediate layers (shown by the decline in Entropy) experience representation collapse for **Pretrained** GPT-2 Small on $5 \times 5$ digit Multiplication (b) and during **Vanilla** training or fine-tuning for both datasets, indicating potential bottlenecks in information flow or feature learning. Tools like **Pause** Goyal et al. (2023b) token-based tuning can't fix it, but our proposed regularization **Seq-VCR** can improve collapse.

Representation collapse in Transformers manifests as a reduction in the diversity of internal representations across layers, particularly in tasks requiring complex reasoning. This collapse hinders the model's ability to capture and process intricate patterns necessary for tasks like multi-digit multiplication. To illustrate this, we analyze the layer-wise representation entropy of a GPT-2 model fine-tuned on the $5 \times 5$ digit multiplication task. As shown in Figure 2(b), there is a significant drop in entropy in the intermediate layers, indicating a collapse in representation diversity. We hypothesize that this representation collapse is the reason why it is very hard for the model to calculate multiplication, carry, and store them, thus resulting in poorer performance, as shown in Figure 6 for the output positions, which require more computation (Figure 1).

We quantify the phenomenon of **representation collapse** in token sequences by computing the matrix-based entropy of representation vectors across transformer model layers. We specifically explore the $\alpha$-order matrix-based entropy (Skean et al., 2023; Giraldo et al., 2014) using a similarity kernel $\kappa$ applied to layer-wise token representations without assuming the underlying distribution. We use a linear kernel defined as $\kappa(a, b) = ab^T$ in our analysis as in (Skean et al., 2024). Assuming, $Z^{(l)} = f_l \in \mathbb{R}^{T \times d}$ be the $l^{th}$ layer representations of $T$ tokens in a sequence with dimension $d$, we compute the pairwise similarity of $T$ token representations to form the Gram matrix $\mathbf{K} = \kappa\left(Z^{(l)}, Z^{(l)}\right) = Z^{(l)} Z^{(l)^T} \in \mathbb{R}^{T \times T}$. Then, we determine the matrix-based entropy of order $\alpha > 0$:

$$S_\alpha\left(Z^l\right) = \frac{1}{1 - \alpha} \log\left[\sum_{i=1}^{T}\left(\frac{\lambda_i(\mathbf{K})}{\text{tr}(\mathbf{K})}\right)^\alpha\right] \tag{2}$$

Equation 2 is similar to the $\alpha$-order Rényi entropy of eigenvalues of the Gram matrix [2], providing insights into their distribution and the underlying token structure. In particular, we employ $\lim_{\alpha \to 1}$, which corresponds to the Shannon or Von Neumann entropy Bach (2022); Boes et al. (2019). Eigenvalues are normalized by the matrix trace $\text{tr}(\mathbf{K}) = \sum_{i=1}^{T} \lambda_i(K)$, then raised to the power $\alpha$, forming

---

[2] $Z^{(l)} Z^{(l)^T}$ (Gram matrix) and $Z^{(l)^T} Z^{(l)}$ (Covariance matrix) have identical nonzero eigenvalues, thus both yield same matrix entropy. When $d < T$, the computation of the covariance matrix is generally more efficient.

a probability distribution over principal components. Each eigenvalue indicates how much variance the corresponding principal component explains (Scholkopf & Smola, 2018). Low entropy results in a heavy-tailed distribution, with a few components dominating, concentrating token information and collapsing the representation space with a lack of representations diversity.

## 3.4 SEQUENTIAL VARIANCE COVARIANCE REGULARIZATION (SEQ-VCR)

To address representation collapse, we introduce Seq-VCR, a regularization technique that enforces high variance and low covariance in intermediate representations of Transformer models. Adapted for sequential data, Seq-VCR builds directly on the variance and covariance regularization principles of VICReg (Bardes et al., 2021) and VCReg (Zhu et al., 2023), with Equation 3 re-purposing the formulations from VICReg to tackle the challenges of sequence modeling.

Seq-VCR is implemented on the final output of the model ($X = f_{cls}$)[3]. The process involves computing a covariance matrix and using a pre-trained model like GPT-2, leading to dimensions such as $f_{cls} \in \mathbb{R}^{T \in \times 50,257}$, which can be computationally challenging. So, for a model like GPT-2, instead of applying regularization on the output of $f_{cls}$, we apply a linear projection that projects the representation of layer $l$ into a more manageable space, expressed as $X = f_{proj}(f_l)$, with $f_{proj} \in \mathbb{R}^{T \times 2048}$. This projection layer is inserted over the representation just before the Seq-VCR regularization loss calculation and is trained exclusively end-to-end with the Seq-VCR loss defined below.

Now we can assume that we have $N$ samples in the batch, so $N$ number of $X$, denoted as $\mathbf{X} \in R^{N \times T \times d}$. We can calculate the covariance matrix across the batch dimension, $C \in R^{T \times d \times d}$.

$$L_{\text{Seq-VCR}} = \frac{1}{T \times d} \sum_{i=1}^{T} \sum_{k=1}^{d} \left( \lambda_1 \underbrace{\max(0, 1 - \sqrt{\mathbf{C}_{i,k,k} + \eta})}_{\text{Variance Term}} + \lambda_2 \underbrace{\sum_{k \neq \hat{k}} (\mathbf{C}_{i,k,\hat{k}})^2}_{\text{Covariance Term}} \right) \tag{3}$$

Where for each position in the sequence, covariance matrix can be calculated across the batch dimension when $\mathbf{X}_{:,i}$ is the input for $i^{th}$ position,

$$C_i(\mathbf{X}_{:,i}) = \frac{1}{N-1} \sum_{j=1}^{N} (\mathbf{X}_{j,i} - \bar{\mathbf{X}}_{:,i})(\mathbf{X}_{j,i} - \bar{\mathbf{X}}_{:,i})^\top, \quad \text{where} \quad \bar{\mathbf{X}}_{:,i} = \frac{1}{N} \sum_{j=1}^{N} \mathbf{X}_{j,i} \tag{4}$$

where $\lambda_1$ and $\lambda_2$ are the coefficients of the variance and covariance regularization terms. $\eta$ is a small constant (set to 0.001 in our experiments), used for numerical stability. Detailed hyper-parameter choices can be found in Appendix A.

The Variance Term encourages unit variance in each dimension, while the Covariance Term penalizes covariance between different dimensions, promoting de-correlation and diversity in representations.

## 3.5 INCORPORATING PAUSE TOKENS

Increasing the model capacity brings significant accuracy boost for solving $n \times n$ digit multiplication tasks (Qiu et al., 2024). While some previous work increases depth to increase the model capacity, an alternative solution is to use pause tokens that serve as explicit indicators for the model to temporarily pause on intermediate states in sequential tasks before proceeding to the next computation step (Goyal et al., 2023b).

Similarly, we enhance the model's reasoning capacity by introducing dummy pause tokens, which act as placeholders for intermediate computation steps. This offers notable benefits over Chain-of-Thought (CoT) reasoning tokens, particularly in reducing inference time (as there are often more

---

[3]Initial tests indicated optimal performance when regularization was utilized on the final layer. Experimentation revealed that additional layers can also be effective. For detailed information on layer-specific regularization, refer to Appendix G.

CoT tokens to decode than pause tokens) and dependency on costly human-supervised data. With pause tokens we can solve tasks like multiplication in a fraction of the time compared to CoT, while performing at a similar accuracy (5 times faster and close to 100% in our experiments. See Appendix E for further details). In all experiments, pause tokens were placed between input and output tokens to emulate CoT reasoning. In particular, the input-output format looked like:

```
<question> </pause_start> <pause> <pause> </pause_end> <answer>.
```

We tried with 2, 4, 6, and 8 pause tokens on $4 \times 4$ and $5 \times 5$ digit multiplication tasks, and we did not find any correlation with task complexity. As such, we used 2 pause tokens in all our experiments. We believe that this may be due to the fact that all pause tokens share the same embedding. Future work will explore the effect of having different embeddings per pause tokens.

### 3.6 TRAINING OBJECTIVE

The overall training objective combines the standard next-token prediction loss with the Seq-VCR regularization. Our final loss **L** is:

$$\mathbf{L} = L_{next} + L_{\text{Seq-VCR}} \tag{5}$$

This encourages the model to not only predict the next token accurately, but also maintain diverse and informative intermediate representations.

## 4 EXPERIMENTS

### 4.1 DATA

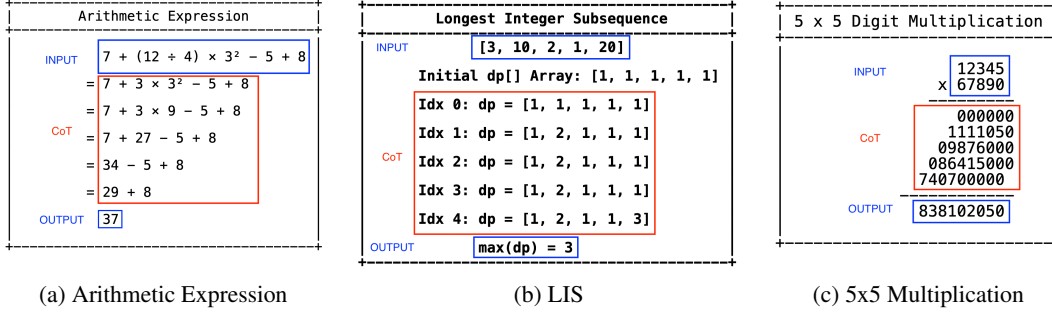

(a) Arithmetic Expression          (b) LIS          (c) 5x5 Multiplication

Figure 3: Illustrations of Input, Output and CoT on the Arithmetic, LIS and mutliplication datasets

We conduct experiments on three tasks: we first consider the **multi-digit multiplication** task from the BIG-bench benchmark (Srivastava et al., 2022), which is the most challenging among arithmetic tasks (Yang et al., 2023). In particular, we use the four-digit ($4 \times 4$) and five-digit ($5 \times 5$) multiplication problems, since these two tasks prove very challenging to solve under no CoT, utilizing the training data generated by Deng et al. (2023).

Next, we focus on the **Arithmetic Expressions** Feng et al. (2024) dataset. This task focuses on evaluating arithmetic expressions. The input sequence for this task is a sequence consisting of numbers, mathematical operators such as addition (+), subtraction (-), multiplication ($\times$), division ($\div$) and brackets, followed by the equal sign. The goal of this task is to calculate the input arithmetic expression and generate the correct result. This task is naturally well-suited for a Chain-of-Thought (CoT) method, where each step does part of the calculation, slowly solving one operation at a time while keeping the other parts the same.

Finally, we also conduct experiments in a more general setting outside arithmetic tasks, called Dynamic Programming (DP). Dynamic Programming is a framework for solving decision-making problems. The core idea in this setting is to break down a complex task into a series of smaller sub-tasks that can be solved sequentially. We chose a very popular problem in this setting called the

**Longest Increasing Sub-sequence (LIS)** as described in the Introduction to Algorithms book (Cormen et al., 2022). For this task, the goal is to find the length of the longest increasing subsequence of a given integer sequence. We generate datasets with different input sequence lengths ranging from {50, 80, 100} as shown in Feng et al. (2024). Moreover, all input sequences, and answers in LIS are bounded-range integers and can therefore be tokenized (similar to the multi-digit multiplication task). The CoT tokens for this task consist of the dynamic programming array plus the final answer. Figure 3 shows a working example for a sample problem in all the datasets considered.

Table 3 in Appendix D provides an overview of the different datasets i.e., Multiplication, Arithmetic Expression and LIS, along with their respective input, chain-of-thought (CoT), and output token details.

## 4.2 MODELS AND TRAINING CONFIGURATIONS

We conduct experiments using two models:

**GPT-2 Small Fine-Tuning** We fine-tune a pre-trained GPT-2 Small model on the multi-digit multiplication tasks. Fine-tuning is performed for 40 epochs with a learning rate of $5 \times 10^{-4}$ and a batch size of 32.

**minGPT Training from Scratch** We train a minGPT model from scratch on the Arithmetic Expressions and LIS datasets. Training is conducted for 100 epochs with a learning rate of $1 \times 10^{-4}$ and a batch size of 128.

We compare five configurations:

- **Vanilla**: Standard training/finetuning without CoT or pause tokens.
- **With CoT**: Training/finetuning with explicit CoT supervision.
- **Pause**: Inserting pause tokens in the input sequence.
- **Seq-VCR**: Applying Seq-VCR regularisation.
- **Seq-VCR + Pause**: Combining Seq-VCR with pause tokens.

## 4.3 RESULTS

### 4.3.1 REPRESENTATION DIVERSITY

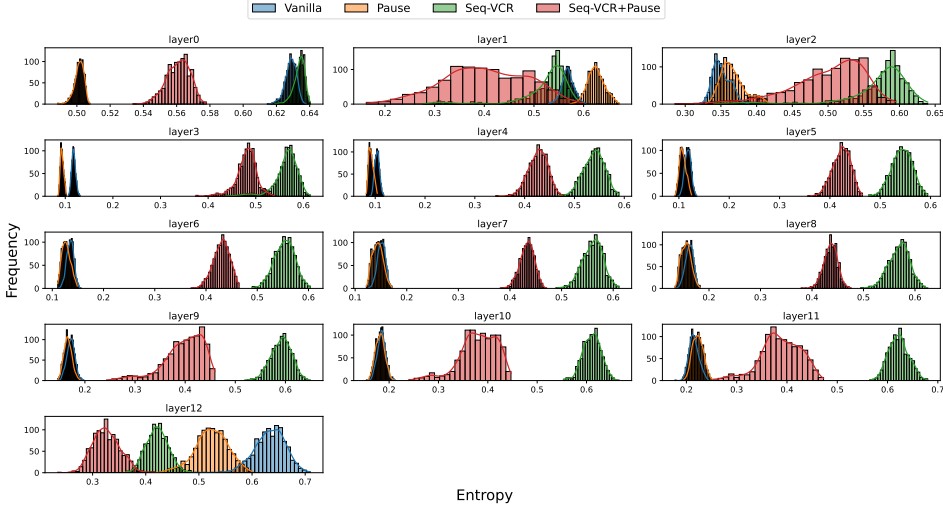

Figure 4: Layer-wise entropy distributions for different configurations on the $5 \times 5$ multiplication task. *Seq-VCR* and *Seq-VCR+Pause* maintain higher entropy across layers, indicating greater representation diversity.

Seq-VCR significantly enhances representation diversity, as evidenced by the layer-wise distribution of representation entropy across different configurations. In Figure 4, we observe that the configurations that employ our regularization technique, particularly *Seq-VCR* and *Seq-VCR+Pause*, exhibit higher entropy values in the intermediate layers compared to the other configurations (*Vanilla*, and *Pause*). This increased diversity in the representation entropy suggests that the model is able to capture a wider range of features and maintain distinct representations. The histograms also reveal that while other configurations show peaks at lower entropy values, indicating potential representation collapse and redundancy in the intermediate representations, our method fosters a more uniform distribution across the layers. This uniformity suggests that the model is using a richer feature space, enabling it to better generalize to unseen data, thus improving performance in downstream tasks.

## 4.4 LEARNING DYNAMICS

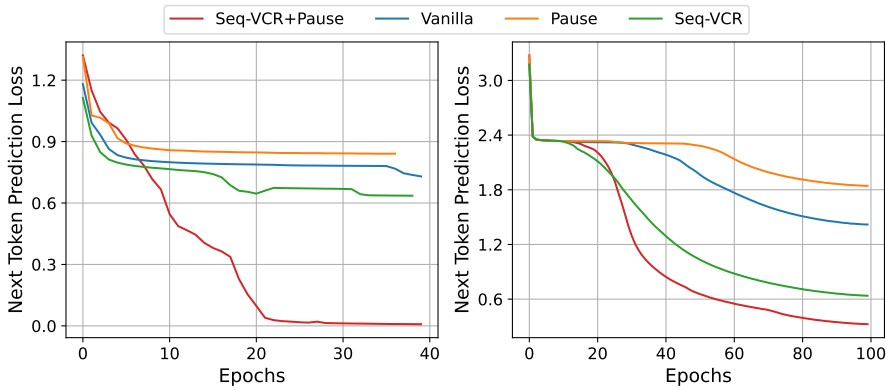

(a) Fine-tuning GPT-2 Small on $5 \times 5$ *digit Multiplication*

(b) Training minGPT from scratch on *Arithmetic Expression*

Figure 5: Learning dynamics (curves for next token prediction loss) illustrating the phase transition observed across datasets when applying our regularization methods. The x-axis represents training epochs, and the y-axis denotes the model's loss. The phase transition is characterized by a sharp reduction in loss, marking a distinct shift in the learning regime when using *Seq-VCR* and *Seq-VCR + Pause*, compared to the gradual decline or saturating curves in other configurations.

Our regularization method, *Seq-VCR* or *Seq-VCR+Pause* cause a significant phase transition (improving the next token prediction loss) in the performance of the model across datasets compared to other configurations such as *Vanilla*, and *Pause*, which do not induce this phase transition as shown in Figure 5. Specifically, this phase transition marks a distinct shift in the model's ability to capture and generalize representations during training or fine-tuning.

Without regularization, learning is difficult and training loss saturates. Seq-VCR improves the learning curve. Adding Seq-VCR with 2 Pause tokens causes a sharp phase transition (Figure 5 (a)) and solves the $5 \times 5$ digit multiplication task. Figure 5 (b) shows similar improvements in the arithmetic reasoning task that is learned from scratch.

## 4.5 RESULTS ON $4 \times 4$ AND $5 \times 5$ DIGITS MULTIPLICATION TASKS

Table 1 shows the results on $4 \times 4$ and $5 \times 5$ digits using a fine-tuned GPT-2 Small. Compared to no CoT configurations like (*Vanilla*, *Pause*), our method enables solving tasks previously not solvable without explicit CoT. We can see the effectiveness of the *Seq-VCR* on $4 \times 4$ digit multiplication task where without pause token it improves performance from $25\%$ to $52\%$ and with 2 pause tokens *Seq-VCR + Pause* achieves accuracy $99.2\%$, which is close to *With CoT* performance and significantly better than other configuration without CoT. It even outperforms the 5-shot prompt(with or without CoT) performance of GPT-3.5 and GPT-4.0.

| Model | Configuration | 4x4 Mult | 5x5 Mult |
|---|---|---|---|
| GPT-3.5 | With CoT | 0.43 | 0.05 |
| | No CoT | 0.02 | 0.00 |
| GPT-4 | With CoT | 0.77 | 0.44 |
| | No CoT | 0.04 | 0.00 |
| GPT-2 Small | With CoT | 1.0 | 1.0 |
| | Vanilla | 0.25 | 0.0 |
| | Pause | 0.28 | 0.0 |
| | Seq-VCR | 0.52 | 0.0 |
| | Seq-VCR + Pause | 0.992 | 0.995 |

Table 1: Accuracy (exact match) on $4 \times 4$ and $5 \times 5$ digits Multiplication Tasks. GPT-3.5 and GPT-4 results are taken from Deng et al. (2024) which are produced by 5-shot prompt

We also see from Table 1 that GPT-4 with CoT 5-shot prompts achieves $44.0\%$ for $5 \times 5$ digit multiplication, while *Vanilla*, *Seq-VCR*, and *Pause* achieve $0\%$ without CoT. Because, all configurations struggle with predicting middle tokens in the output sequence (shown in Figure 6). This happens because those middle tokens require more computations to calculate successive carry and store them, since these tokens require more computation (as shown in Figure 1), thus making them difficult sub-tasks than predicting the peripheral tokens which require less computations, thus easier to solve. However, *Seq-VCR+Pause* with 2 Pause tokens solves the $5 \times 5$ digit multiplication task, previously unsolved without CoT tokens. Figure 2(b) shows our regularization *Seq-VCR* increases intermediate layer entropy, allowing more exploration in the representation space. We hypothesize that along with better representation by *Seq-VCR*, 2 Pause tokens increase the computational ability, allowing the model to carry the necessary multiplication values and solve the task. More results by training GPT-2 from scratch are in the Appendix H.

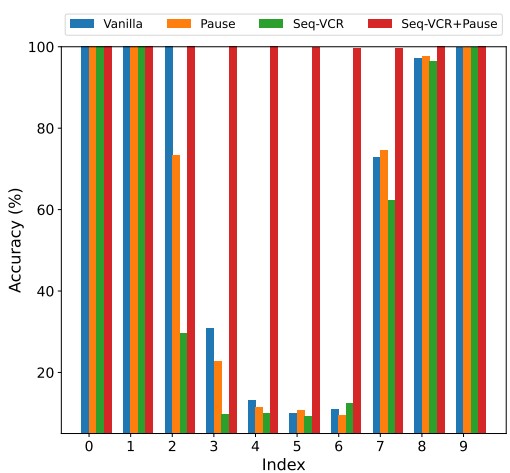

Figure 6: Position-wise accuracy of Multiplication of 5x5 digits with different configuration of GPT-2 Small. We see that models fail on the middle tokens as they require more compute.

### 4.6 RESULTS ON ARITHMETIC EXPRESSION AND LONGEST INCREASING SUB-SEQUENCE (LIS) DATASETS

Figure 7 illustrates how the accuracy varies among different methods (represented by different bar colors in the plot) and with changing levels of task complexity, such as the number of operators in the Arithmetic Expressions Dataset or the input sequence length in the case of the LIS dataset.

We can observe that as the task complexity is low, i.e. in case of 4 operators in the arithmetic expression and input sequence length of 50 in the LIS dataset, all the methods perform quite well. But, as the number of operators or sequence length increases, the tasks become more complex, which shows the need for models other than the vanilla baselines.

Seq-VCR demonstrates substantial improvement over vanilla models and achieved results comparable to the chain-of-thought (CoT) prompting method as shown in Feng et al. (2024). Although CoT outperforms the other methods, there is an increase in the serial computation due to its step-by-step reasoning approach. On the other hand, Pause tokens and regularization primarily boost parallel computation, enhancing the model's expressivity for these types of problem without significantly increasing serial computation. The results presented in Figure 7 are based on models with a 5-layer configuration, optimized using the best-performing number of pause tokens. We find that the number of pause tokens that works best for a particular model and layer config vary a bit. For example,

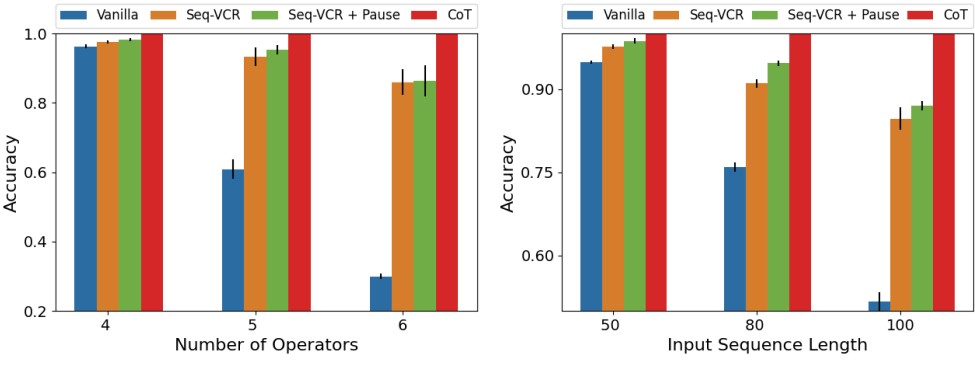

(a) Test accuracy on Arithmetic Expressions Dataset    (b) Test accuracy on LIS Dataset

Figure 7: Performance of each method across tasks of varying difficulty

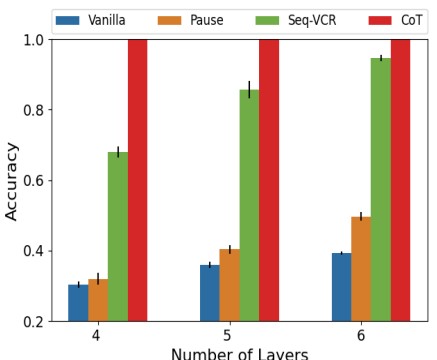
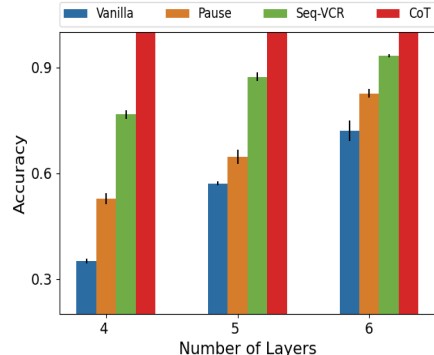

(a) Test accuracy on 6 operator Arithmetic Expression

(b) Test accuracy on LIS Dataset with 100 Input Sequence Length

Figure 8: Performance of each method on the complex task w/ varying number of layers

the best working Seq-VCR + Pause model uses 7 pause tokens in the case of the 5 operator task in Arithmetic Expression, while 4 pause tokens work the best for the LIS task with 80 as input sequence length. In Figure 8 we show how the different models perform in the most complex task in each dataset. We also clearly see that more depth of the model helps every method in solving the task better. Another conclusion we can draw from Figures 7 and 8 is that although the combination of Seq-VCR with Pause Tokens works best (almost close to the CoT method), Seq-VCR achieves a similar test accuracy without the use of Pause Tokens in both datasets. All results in Figures 7 and 8 are over 3 seed runs. Detailed ablation studies, exploring variations in the number of pause tokens, layers, and task complexities, can be found in the Appendices B and C.

## 5   CONCLUSION

Despite the remarkable success of large transformer based language models, they exhibit deficiencies in multi-step reasoning tasks. One proposed remedy is the application of Chain-of-Thought supervision, which, although effective, entails considerable cost. Conversely, there has been limited exploration into enhancing the capacity of the pre-trained model representation itself without explicit supervision. Our analysis indicates that intermediate layer representation collapse detrimentally affects the computation of intermediate information essential for solving arithmetic reasoning tasks. Our proposed regularization technique enhances the model's representation capacity, thereby improving its ability to solve these tasks. We think that further research is required to enhance the capability of pre-training as well, where improved representation learning will enhance the model with greater capabilities.

ACKNOWLEDGMENTS

We acknowledge the support of the Canada CIFAR AI Chair Program and IVADO. We thank Mila and Compute Canada for providing computational resources.

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

## A  HYPERPARAMETERS

| Parameter | Value/Details |
|---|---|
| Learning Rate | 0.0001 |
| Batch Size | 128 |
| Optimizer | AdamW |
| Dropout | 0.1 |
| Attn. Heads | 4 |
| Epochs | 100 |
| Number of Layers | 4, 5, 6 |
| Input Sequence Length | 50, 80, 100 |
| # of Operators | 4, 5, 6 |
| Total Compute Resources | 1 32GB GPU, 6CPU, 32 GB RAM |
| Total Job Time | max 24 Hrs |

Table 2: Summary of Hyper-parameters, Compute Resources, and Time for Experiments. Each experiment was run on same configurations of GPUs, with the total time dependent on the number of layers and input parameters such as Dataset, task complexity.

We manually searched for hyperparameters, we did not do exhaustive searches such as grid search or random search. We found this was enough to find good hyperparameters for our tasks. For the two coefficients $\lambda_1$ and $\lambda_2$ in Equation **??**, we keep their proportion similar to(Bardes et al., 2021). For multiplication tasks $\lambda_1 = 1.0$ and $\lambda_2 = 0.004$ and for other tasks we use $\lambda_1 = 0.1$ and $\lambda_2 = 0.5$. For computing the covariance matrix, we use a batch size of 32 for multiplication tasks and 128 for others. Other hyperparams like learning rate and batch size values are similar to other related works (Deng et al., 2023; Feng et al., 2024).

Regarding the number of pause tokens, we tried 2, 4, 6, 8 pause tokens on 4x4 and 5x5 digit multiplication tasks, and we did not find any correlation with task complexity. We believe it may be due to the fact that all the pause tokens share the same embedding.

## B  MODEL COMPLEXITY VS TASK DIFFICULTY

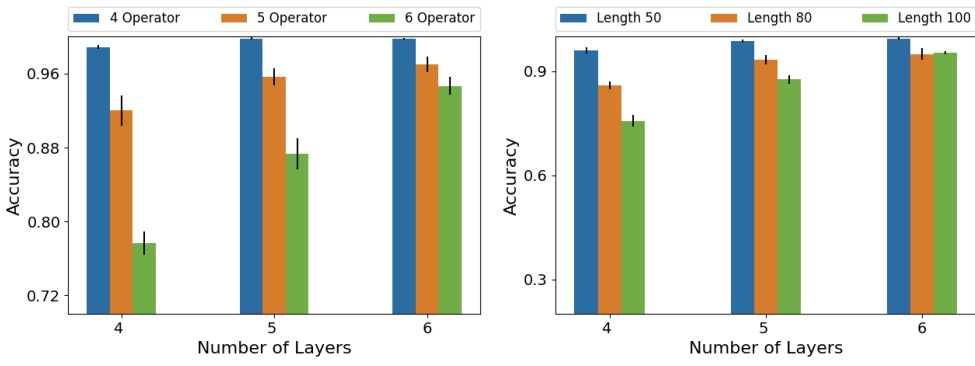

(a) Test accuracy on Arithmetic Expressions Dataset    (b) Test accuracy on LIS Dataset

Figure 9: Seq-VCR + Pause Accuracy Vs Number of Layers for tasks of different complexity in Arithmetic Expressions and LIS

Both plots in Figure9 demonstrate the trade-offs between model complexity (in terms of layers) and the complexity of the input (number of operators or sequence length). The model performs best when handling simpler configurations, such as fewer operators or shorter sequences, and tends to struggle with more complex setups as the number of operators or sequence length increases. Moreover, Increasing the number of layers seems to mitigate performance drops to some extent, especially in simpler cases, but it cannot fully offset the challenges posed by more complex inputs. All results in figure 9 are over 3 seed runs.

## C   NUMBER OF PAUSE TOKENS VS TASK COMPLEXITY

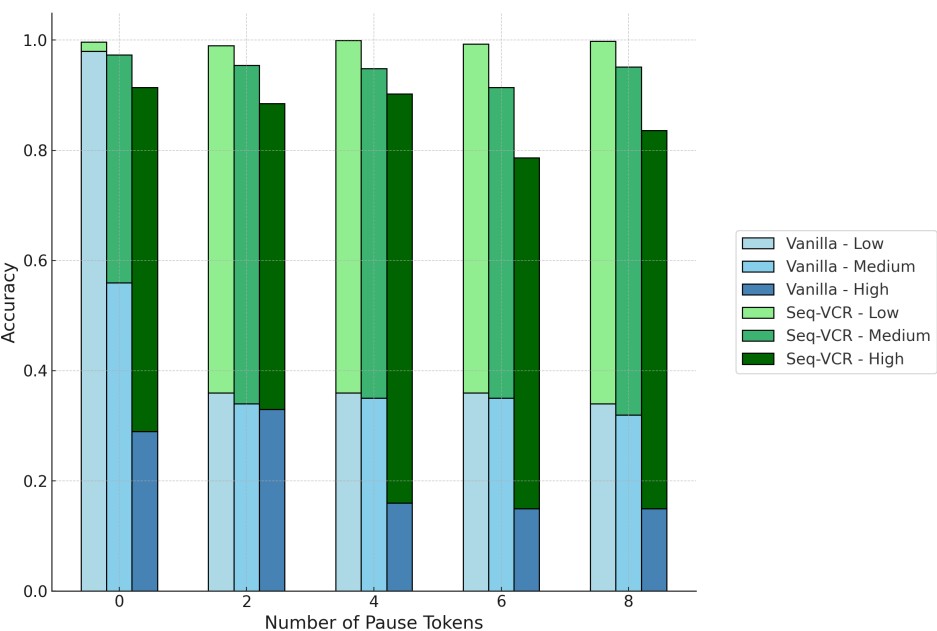

Figure 10: Ablation of Varying Pause Tokens and Comparing Vanilla Vs Seq-VCR + Pause Tokens for different Task Complexities in the arithmetic dataset. Low, High, Medium refer to 4, 5, 6 arithmetic Operators respectively, when number of layers is fixed to 5.

Figure10 shows that Seq-VCR + Pause model consistently achieves high accuracy (around 0.8 or higher) across all numbers of pause tokens, indicating that the addition of pause tokens does not affect its performance. The Vanilla model, in contrast, exhibits much lower accuracy across all settings except for the low-complexity task with four operators. We hypothesize that this is because five layers are likely sufficient to solve such a simple task.

For this task, where the Vanilla model performs better, we observe a notable drop in accuracy when pause tokens are added. We speculate that this occurs because, in scenarios where the model has sufficient capacity to solve the task, pause tokens introduce distractions that hinder performance rather than enhancing it. However, in tasks with insufficient depth, such as the 5x5 digit multiplication task on GPT2-Small, pause tokens prove beneficial, as shown in Table 1.

# D   DATA STATISTICS

| Dataset | Task | Size | # Input Tokens | # CoT Tokens | # Output Tokens |
|---|---|---|---|---|---|
| Multiplication | 4 x 4 Mult | 808k | 9 | 47 | 8 |
| | 5 x 5 Mult | 808k | 11 | 75 | 10 |
| Arithmetic Expression | 4 Operators | 1M | 19 | 24 | 1 |
| | 5 Operators | 1M | 23 | 40 | 1 |
| | 6 Operators | 1M | 27 | 60 | 1 |
| LIS | Seq Length 50 | 1M | 50 | 50 | 1 |
| | Seq Length 80 | 1M | 80 | 80 | 1 |
| | Seq Length 100 | 1M | 100 | 100 | 1 |

Table 3: Dataset statistics. Size refers to the training set. The number of input, output, and intermediate chain of thought tokens are median values on the validation set. The number of tokens are based on the GPT-2 tokenizer, and a special ending symbol is counted for both intermediate tokens and output tokens.

# E  SPEEDUP AND ACCURACY TRADE-OFF FOR PAUSE AND CoT TOKENS

Seq-VCR offers notable benefits over Chain-of-Thought (CoT) reasoning, particularly in reducing inference time and dependency on costly human-supervised data. Unlike CoT, which requires extensive labeled data for multi-step reasoning, Seq-VCR uses a few dummy pause tokens to solve tasks like multiplication in a fraction of the time (5 times faster), while performing at a similar accuracy close to 100% in our experiments (see table below). Seq-VCR's efficiency in both inference time, data requirements, and accuracy makes it a more scalable and robust approach compared to CoT.

To compute the normalized throughput for inference, we use the following equation, as in the Deng et al. (2024) paper:

$$T_{\text{norm}} = \frac{T_{\text{target}}}{T_{\text{base}}}$$

Here:

- $T_{\text{norm}}$ is the normalized throughput, which represents the relative inference speed.
- $T_{\text{target}}$ is the throughput (number of examples processed per second) when using target method.
- $T_{\text{base}}$ is the throughput (number of examples processed per second) for the baseline model without Chain of Thought or Pause tokens.

| Method | $\mathbf{T}_{norm}$ 4x4 Mult | $\mathbf{T}_{norm}$ 5x5 Mult | Accuracy 4x4 Mult | Accuracy 5x5 Mult |
|---|---|---|---|---|
| No CoT | 1.0 | 1.0 | 0.25 | 0.0 |
| With CoT | 0.17 | 0.14 | 1.0 | 1.0 |
| Seq-VCR + Pause (2) | 0.95 | 0.91 | 0.992 | 0.995 |

Table 4: Normalized Throughput (the higher the better) and Accuracy measures on $4 \times 4$ and $5 \times 5$ digit multiplication without CoT tokens, with CoT tokens, and with 2 pause tokens.

Note, that it is expected that models with CoT tokens perform the best as CoT tokens carry more useful information than simple dummy pause tokens. However, they are more expensive to get (human labor) and require more compute to generate at inference time (since there are more of them). In that sense, Seq-VCR will always scale better than CoT, no matter the model size.

# F  COMPUTATION OF THE COVARIANCE MATRIX ACROSS BOTH THE BATCH AND LENGTH DIMENSIONS

Here we present an ablation study done by fine-tuning GPT2-small on the $5 \times 5$ digit multiplication. We compared the performance of calculating the covariance matrix across the batch dimension vs both the batch and the length dimensions and present results in the Table below.

| GPT2-small (fine-tuned) | Accuracy | Compute Complexity |
|---|---|---|
| baseline | 0 | 0 |
| Seq-VCR - **batch** dim | 0.99 | $\mathcal{O}(b \cdot d^2)$ |
| Seq-VCR - **batch+length** dim | 0.98 | $\mathcal{O}(b \cdot n \cdot d^2)$ |

Table 5: Accuracy and compute complexity (when b, n and d are batch size, # tokens and feature dimension respectively) difference between computing the covariance matrix across the batch vs length+batch dimensions.

We find that there are no significant differences (between 98 and 99% accuracy on 5x5 digit multiplication). However the computation time is $n$ times larger on average when using both the batch and length dimensions, which grows with the increase of sequence lengths.

# G    ABALATION OF LAYERS

We apply Seq-VCR to all layers while fine-tuning GPT-2-small on a 5x5 multiplication task. The results indicate improved performance when Seq-VCR is applied to the last layer.

| Layer | L0 | L11 | L12 |
|---|---|---|---|
| Accuracy | 0.97 | 0.98 | 0.99 |

Table 6: Performance of Seq-VCR + Pause across different layers.

With $\lambda_1 = 1.0$, $\lambda_2 = 0.004$, and pause = 2, we achieve these results, where applying Seq-VCR on the first and last two layers almost solves the task, but for the rest of the layers it does not work. However, hyperparameter tuning may be required for other layers.

# H    PRE-TRAINED MODELS VS TRAINING FROM SCRATCH

Here, we run more experiments to emphasize the advantage of fine-tuning pre-trained models. We train a GPT2-small model from scratch on $5 \times 5$ digit multiplication tasks over multiple hyperparameter settings and present the results in the table below:

| GPT2-small | From Scratch | | Fine-Tuned | |
|---|---|---|---|---|
| Layer | $L12$ | $L0$ | $L12$ | $L0$ |
| Baseline | 0 | 0 | 0 | 0 |
| Seq-VCR - **batch** dim | 0.03 | 0.97 | 0.99 | 0.97 |
| Seq-VCR - **batch+length** dim | 0.87 | 0.99 | 0.98 | 1.0 |

Table 7: Accuracy of pre-trained and training from scratch by computing the covariance matrix across the batch vs. length+batch dimensions, applying Seq-VCR on 1st (L0) and last (L12) layers

Training models from scratch is more sensitive to hyperparameter choices, such as batch size, and performs poorly when Seq-VCR is applied to the last layer. However, computing covariance across both the batch and length dimensions significantly improves performance. In contrast, fine-tuning GPT-2 proves to be more stable than training from scratch. Interestingly, as shown in Table 7, applying regularization to the 1st(L0) layer is effective for both fine-tuning and training from scratch. This is likely because Seq-VCR at the 1st(L0) layer enhances collapse prevention, as shown in Figure 15.

# I    EXPERIMENTS WITH LARGER MODEL

To test the effectiveness of our approach on larger, more recent models we applied our Sec-VCR regularization method during fine-tuning of a Llama3.2-1B model with LoRA($r = 64$) adapters on the challenging 5x5 digit multiplication task.

We measure the representation collapse across layers for the finetuned model with and without our regularization and report results in Figure 11 below.

| LLama 3.2-1B | Accuracy |
|---|---|
| Vanilla | 0 |
| Seq-VCR | 0.974 |

Table 8: Accuracy of finetuning Llama3.2-1B model on 5x5 digit multiplication task

We find that the model trained with Sec-VCR has a higher average entropy (reduced collapse) by 10-20% compared to the baseline llama model and achieving 97.4% (Table 8) accuracy. This experiment highlights that collapse is not an isolated case and is still an issue in more recent models.

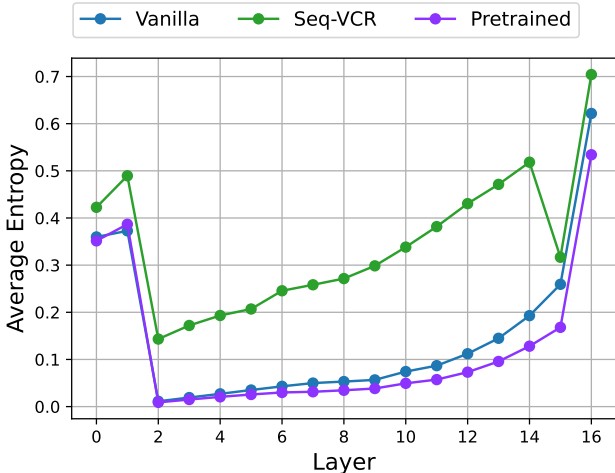

Figure 11: Representation collapse across llama3.2-1B layers on the $5 \times 5$ digit multiplication task.

## J    COLLAPSE AT SCALE

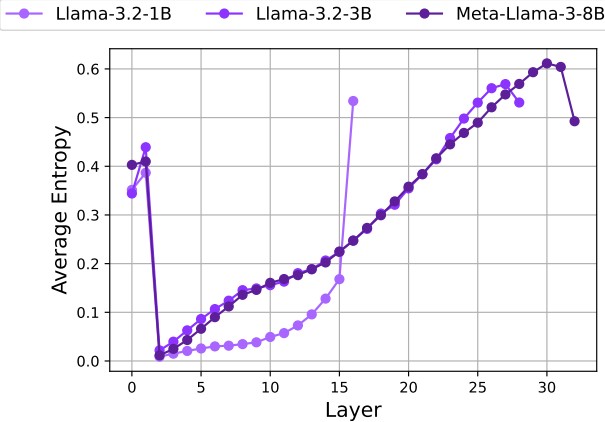

Figure 12: Representation collapse across scale (1B, 3B and 8B) of LLama pretrained models

We tested pre-trained Llama models of varying scales on the 5x5 digit multiplication dataset and observed consistent representation collapse across all model sizes (Figure 12). This highlights that the issue persists regardless of scale, emphasizing the need for effective regularization techniques like Seq-VCR to mitigate collapse and improve intermediate reasoning capabilities.

## K    COLLAPSE MITIGATION IN OTHER BENCHMARKS

To test the effectiveness of our method to other domains, we finetuned a Code-GPT-2 Small model on the CodeXGLUE-text-to-code benchmark[4]. We measure the representation collapse across layers for the finetuned model with and without our regularization and report results in Figure 13 below.

We find that the model trained with Sec-VCR has a higher average entropy (reduced collapse) than the baseline pre-trained llama model by 3-4%. This experiment showcases that our method also reduces collapse in other domains.

---

[4]https://github.com/microsoft/CodeXGLUE/tree/main/Text-Code/text-to-code

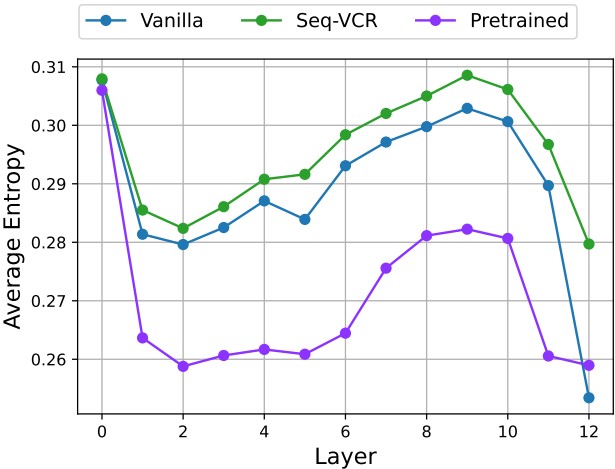

Figure 13: Representation collapse across Code-GPT-2 Small layers on the CodeXGLUE-text-to-code task.

## L  COLLAPSE MITIGATION DURING PRE-TRAINING

To evaluate the effectiveness of our method in mitigating representation collapse during pre-training, we trained GPT-2-small from scratch on the C4 dataset. The regularizer was applied either to the last layer (L12) or the first layer (L0). As shown in Figure 14, the validation perplexity achieved by our method is similar to the baseline Vanilla validation perplexity. For $L0$, $\lambda_1 = 1.0$ and $\lambda_2 = 0.01$ and for $L12$, $\lambda_1 = 0.5$ and $\lambda_2 = 0.1$

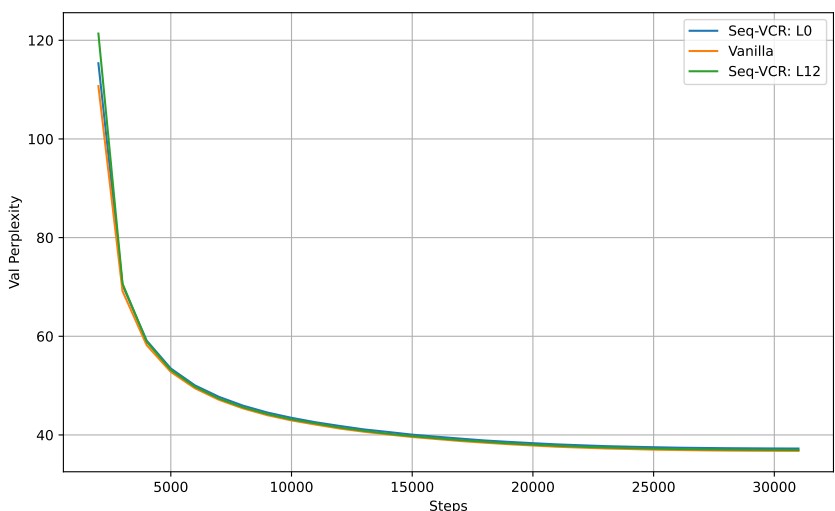

Figure 14: Validation Perplexity of GPT-2 Small trained on the C4 dataset.

However, Figure 15 demonstrates that Seq-VCR effectively mitigates collapse across layers. We find that the model trained with Sec-VCR has a higher average entropy (reduced collapse) than the vanila trained model by 3-5%. This experiment showcases that our method also reduces collapse in other domains. Notably, applying the regularizer to the first layer resulted in a more stable prevention of collapse in intermediate layers.

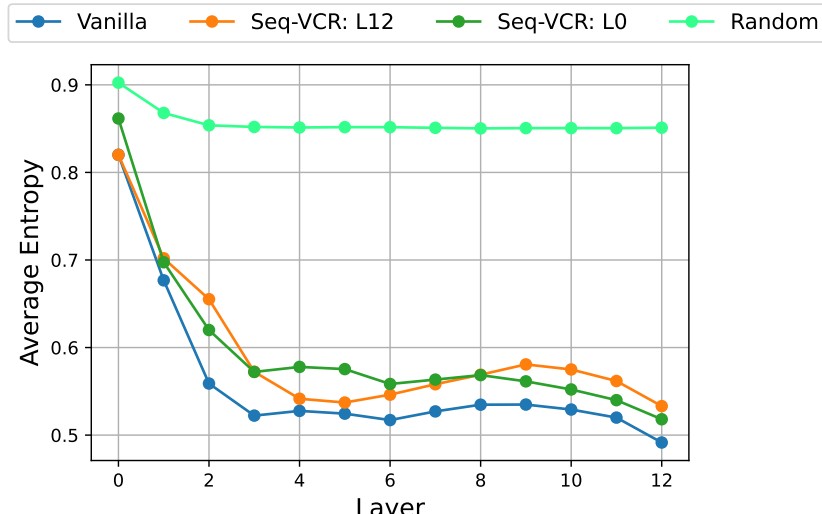

Figure 15: Representation collapse across GPT-2 Small layers after the C4 pretraining.

## M    RESULTS ON GSM8K

We performed further experiments to fine-tune the GPT-2 Small model using an enhanced version of the GSM8K dataset[1], incorporating and excluding Seq-VCR, Pause (2), and CoT. The findings are presented below.

| Method | Accuracy |
|---|---|
| Vanilla | 0.191 |
| CoT | 0.437 |
| Pause(2) | 0.197 |
| Seq-VCR | 0.198 |
| Seq-VCR + Pause(2) | 0.202 |

Table 9: Comparison of Various Methods by Fine-tuning GPT-2 Small on the GSM8K Dataset.

For this dataset, performance improves slightly compared to Vanilla fine-tuning when using Seq-VCR without pauses, and even more when applying pauses with Seq-VCR.

