# OpenReview forum: "Seq-VCR: Preventing  Collapse in Intermediate Transformer Representations for Enhanced Reasoning"
_ICLR.cc/2025/Conference — ICLR 2025 Poster_

### Official Review · Reviewer_LszK · 2024-11-02

**Soundness:** 2
**Presentation:** 3
**Contribution:** 2
**Rating:** 5
**Confidence:** 4

**Summary:**

This work identifies representation collapse in the LLM intermediate layers as a key factor limiting their arithmetic reasoning capabilities. The paper proposes sequential variance-covariance regularization (Seq-VCR). It then combines Seq-VCR with pause tokens to prevent the representation collapse. Experiments on GPT-2-small and minGPT demonstrate the effectiveness in improving accuracy on arithmetic reasoning.

**Strengths:**

1. The identified representation collapse is quite interesting

2. The proposed method, including Seq-VCR regularization loss and pause tokens, demonstrates novelty and effectiveness based on the experimental results.

**Weaknesses:**

1. The representation collapse experiment was conducted only on GPT-2. I am curious whether this phenomenon occurs in more recent and larger LLMs, such as LLaMA 3 or LLaMA 3.1. The authors should either include additional experiments or provide a theoretical analysis to demonstrate that this is not an isolated case.

2. While the proposed Seq-VCR regularization loss has been shown to be effective in arithmetic reasoning tasks, I wonder whether adding this loss after the next token prediction loss would impact the LLM's performance on other tasks  (e.g., math reasoning and general MMLU). If it does have an effect, then this method may not be widely applicable. I encourage the authors to discuss this point.

**Questions:**

see the weaknesses section.

---

> ### Author Response · Authors · 2024-11-23
>
> We would like to thank the reviewer for the time and valuable feedback. Below, we address the concerns raised.
>
> **1) Representation Collapse on larger LLMs:**
> We have conducted additional large scale experiments and scaling analysis of collapse of different size of LLama models based on the reviewer's feedback. Please check the Generic Response (1) for more details.
>
> **2) Expanding Seq-VCR to broader language tasks:**
> We appreciate the opportunity to address this valid concern. In response, we conducted additional experiments, which along with other details are discussed in the Generic Response. Please refer to Generic Response (2) for further details.
>
> We hope we addressed all your concerns and please let us know if anything else is unclear. We would like to request the reviewer to consider increasing the score.

---

> > ### Comment · Reviewer_LszK · 2024-11-27
> > **Thank you for your rebuttal**
> >
> > Thank you for the additional experiments on LLaMA 3. Regarding the second question, instead of showing the entropy presented in Figure 13, could you provide the standard evaluation on general benchmarks, such as MMLU/GSM8K accuracy?

---

> > > ### Author Response · Authors · 2024-12-01
> > >
> > > Dear Reviewer LszK,
> > >
> > > As we approach the end of the discussion period, we want to ensure our responses have addressed your comments and kindly request your consideration in revising the rating.

---

> ### Author Response · Authors · 2024-11-29
>
> **Could you provide the standard evaluation on general benchmarks, such as MMLU/GSM8K accuracy?**
>
> We would like to thank the reviewer for their feedback. We conducted additional experiments to fine-tune the GPT-2 Small model on an augmented version of the GSM8K dataset[1], both with and without Seq-VCR, Pause (2), and CoT. The results are shown below.
>
> | Method               | Value  |
> |----------------------|--------|
> | Vanilla              | 0.191  |
> | CoT                  | 0.437  |
> | Pause(2)            | 0.197  |
> | Seq-VCR             | 0.198  |
> | Seq-VCR + Pause(2)  | 0.202  |
>
> For this dataset, we observe slight performance improvement when applying pauses, Seq-VCR without Pause, and slightly more performance improvement when using Seq-VCR with pauses.
>
> In addition to GSM8k, we have also included some preliminary pretraining results in Appendix L by training GPT2-Small on the **C4 dataset**. Seq-VCR models maintain performance on validation perplexity compared to the vanilla baseline model, while still increasing representation entropy (hence reducing collapse).
>
> In summary, these two additional experiments show that our method:
>
> - Does not penalize the performance on more generic language tasks
>
> - Increases representation entropy (reduces representation collapse) in all cases
>
> - Improves performance on mathematical reasoning tasks
>
> [1] Deng, Yuntian, et al. "Implicit chain of thought reasoning via knowledge distillation." arXiv preprint arXiv:2311.01460 (2023).

---

### Official Review · Reviewer_YSNL · 2024-11-02

**Soundness:** 2
**Presentation:** 1
**Contribution:** 4
**Rating:** 6
**Confidence:** 5

**Summary:**

Background: variance-covariance regularization (VICReg/VCReg) is a technique that was pioneered in vision models.  Given a batch of inputs, the technique uses an NN to encode those inputs to a batch of embedding vectors, and then computes a covariance matrix for the embedding vectors.  It introduces two losses based the covariance matrix: (a) the variance loss ensures that every dimension of the embedding vector has different values, distributed across the batch, and (b) the covariance loss ensures that different dimensions are not correlated.  In vision models, these two losses guard against representational collapse.

The authors of this paper adapt VICReg from the vision domain to transformer-based language models.  They show that when combined with pause tokens, VICReg (now renamed to Seq-VCR) produces large improvements in several tasks that LLMs are usually very bad on -- multidigit arithmetic, arithmetic expressions, and a longest-increasing-subsequence task.

**Strengths:**

The application of VICReg to language models is novel as far as I know, and the experimental results are very compelling.  This could potentially be a high-impact paper in improving the reasoning capabilities of LLMs.

**Weaknesses:**

Unfortunately, the paper itself is hastily and sloppily written, and difficult to follow in places.  I had numerous questions when reading it that are not addressed by the text.  The current draft does not contain all of the information necessary to replicate these experiments, and does not discuss many of the crucial design decisions.  The authors claim that there is an "Appendix A", but neglected to provide any supplementary material.  See "questions" section below for specific questions.

One of the author's central claims is that transformers suffer from representational collapse, but I do not think that they adequately make that point based on the experimental evidence.  There are only two entropy charts in Figure 2, which cover only two narrow tasks.  On one of those charts (a) the collapse seems minimal at best, while on the other (b) the addition of pause tokens (the second key technique that the authors propose) actually increases collapse, rather than decreasing it.  I would need to see a much larger set of studies, over a variety of different tasks, including general language modeling tasks (translation etc.) to fully buy the author's argument about collapse.  If the authors did such a study, however, it would be a significant breakthrough.

Similarly, I would like to know what the effects of VICReg are on more general language modeling tasks.  If the technique helps the model multiply 5-digit numbers after fine-tuning, but otherwise degrades peformance on most other language modeling tasks, then the technique is useless.  Because the authors do not perform this ablation, it is impossible for me to evaluate whether this is a high-impact advance over SOTA, or a trivial result.

Finally, the use of pause tokens is interesting, but also seems haphazard.  They authors themselves admit that the number of pause tokens is task-specific.  To employ this technique more widely, I would need to see a more comprehensive test of where, how many, and under what circumstances pause tokens should be added.

More specific criticisms:

Equation (3) defines the Seq-VCR loss.  The text of the paper claims that it is "inspired by" prior work, and cites such work appropriately, but it is more than just "inspired".  Equation (3) is lifted almost verbatim from the orginal VICReg (Bardes 2021) and VCReg (Zhu 2023) papers, and the authors need to be crystal clear about the source of that equation.

(As a minor nit, it is unclear to me whether or not the covariance term in equation (3) should have an additional 1/(d-1) factor; VICReg has the term, while VCReg does not.  I would have appreciated it if the authors explained why they chose one version over the other.)

For further clarity, the authors should also devote a few lines to defining how the covariance matrix C is computed; as is done in other papers.  Otherwise, it can easily be confused with the cross-correlation matrix of the Barlow twins technique, which the authors also cite as inspiration.

**Questions:**

(1) Pause tokens are a crucial part of the author's technique, but at no point do the authors describe where, and how, the pause tokens are added to the input.

(2) Representational collapsed supposedly happens in the intermediate layers of the transformer, and yet the Lseq-VCR loss term is only applied to the final layer.  (Line 225).  Shouldn't it be applied to the intermediate layers, where you measure the entropy?  Why not?

(3) Equation (3) introduces $\lambda_1$ and $\lambda_2$ as hyperparameters, but the paper fails to say what they are set to.

(4) What batch size is used for computing the covariance matrix?

(5) Equation 3 computes the covariance matrix only across the batch dimension.  Why?  In a transformer, you could potentially use the length dimension as well, which would drastically increase the effective batch size.  Did you do an ablation study which showed that to be ineffective?

(6) How is the projection layer $f_{proj}$ trained?

(7) For GPT2 on multiplication, you fine-tune a pre-trained GPT2 model, despite the fact that the pre-trained GPT2 has no real multiplication ability to start with.  Why bother with a pre-trained model, instead of just training from scratch, as you do with minGPT on arithmetic expressions?

---

> ### Author Response · Authors · 2024-11-24
>
> Thank you very much for the detailed review and feedback. We address all your concerns below and will clarify in the paper accordingly. Please let us know if anything is still unclear.
>
> -**Appendix included in the main text:**
>
> We have included all supplementary materials directly in the main PDF document. We did not split it into two separate PDFs, as this was not explicitly required by the ICLR guidelines, and we do not have large zip files to share. The revised document contains several additional appendix sections.
>
> -**About representational collapse & the effects of Seq-VCR on more general language modelling tasks:**
>
> Please check the generic response section (1)
>
> -**About pause tokens (Q1):**
>
> Please check the generic response section (2)
>
> -**Regularization on intermediate layers (Q2):**
>
> In our initial experiments, we applied the regularization across all layers and observed better performance when it was applied only to the last layer. We think this is because, when applied to the last layer, the regularization loss gradients update all layers, whereas applying it to intermediate layers only updates those layers. While we found that applying it to the last layer works best, other layers could potentially yield similar results. We have added this clarification as a footnote in Section 3.4 of the manuscript, highlighted in blue-colored text.
>
> -**Hyperparameters (Q3, Q4):**
>
> For multiplication tasks, we set $\lambda_1 = 1.0$ and $\lambda_2 = 0.004$, while for other tasks, we use $\lambda_1 = 0.1$ and $\lambda_2 = 0.5$.
>
> For computing the covariance matrix, we use a batch size of 32 for multiplication tasks and 128 for other tasks. These details have been added to Appendix A of the revised manuscript.
>
> -**Computation of the covariance matrix across both the batch and length dimensions (Q5):**
>
>  **New experiments:**
> Thank you for pointing this out. We indeed missed this ablation when increasing the effective batch size. Based on your comment, we conducted an ablation study on the 5x5 digit multiplication task by fine-tuning GPT2-small. We compared the performance of calculating the covariance matrix across the batch dimension alone vs. across both the batch and length dimensions.
>
> **Results:**  We found that there are no significant differences in accuracy across multiple hyperparameter settings (ranging from 98% to 99% on the 5x5 digit multiplication task). However, the computation time is, on average, n times larger (depending on the sequence length, n) when calculating the covariance matrix across both the batch and length dimensions. Full results can be found in Appendix F.
>
> -**How projection layer $f_{proj}$, is trained (Q6):**
>
> The projection layer, $f_{proj}$, is added over the representation before calculating the loss over the projected representations. It is trained end-to-end exclusively using the Seq-VCR loss (Equation 3). We clarified this point in Section 3.4 of the paper in blue-colored text.
>
> -**Pre-trained models vs training from scratch (Q7):**
>
> **New experiments:**
> Based on your comment, we run more experiments to emphasize the advantage of training pre-trained models. We trained a gpt2 model from scratch on 5x5 digit multiplication tasks over multiple hyperparameter settings.
>
> **Results:**
> Training models from scratch is more sensitive to hyperparameter choices such as batch size, hence your suggested way of increasing effective batch size worked better. For the full analysis, please check Table 6 in Appendix G.
>
> -**About equation 3:**
>
> The main contribution is to apply this regularization in Transformers on text-domain as the original idea was proposed for the image representation learning. We acknowledge that the equation is the same as VICReg[2] and therefore we use the same normalizing factors as in VICReg.
>
> --**Covariance Matrix Computation:**
>
> We computed the covariance matrix, C  based on equation 3 in VICREG [2], so it's not the cross-correlation matrix as in the Barlow twins.
>
> [1] Goyal, Sachin, et al. "Think before you speak: Training language models with pause tokens." arXiv preprint arXiv:2310.02226 (2023).
>
> [2]  Bardes, Adrien, Jean Ponce, and Yann LeCun. "Vicreg: Variance-invariance-covariance regularization for self-supervised learning." arXiv preprint arXiv:2105.04906 (2021).
>
> We hope we have thoroughly addressed all your concerns and clarified any ambiguities. We kindly request the reviewer to consider revisiting the evaluation and potentially increasing the score based on these updates.

---

> ### Comment · Reviewer_YSNL · 2024-11-26
>
> I would like to thank the authors for responding to my questions, running additional experiments, and for updating the paper.  I especially appreciate the fact that modifications to the paper were done in blue, which makes it easy to see what was changed.
>
> **(1)** Unfortunately, some of my original criticisms have not been addressed.  The most glaring weakness of the paper, by far, is the fact that it does not measure the effect of SeqVCR on general-purpose language modeling.  5x5 digit multiplication problems are an extremely narrow downstream task, and a technique which improves performance on that task, but only at the expense of general language modeling, is useless.  To clarify, I do not expect the authors to pre-train a large-scale LLM!  It would be sufficient to train a small (e.g. 100M parameter) model from scratch on a general dataset like C4, and measure perplexity for next-token prediction.  The purpose of the experiment would be merely to show that VCReg does not hurt performance on general LM tasks -- if the perplexity is the same, then your technique is a winner.
>
> **(2)** A second weakness of the paper is that it conflates the effects of two completely different techniques -- SeqVCR and pause tokens.  For the 5x5 digit multiplication task, the two seem to work together synergistically.  However, in Appendix C. Figure 10, the best performing model seems to be the one with no pause tokens at all!  And pause tokens seem to be particularly harmful to the vanilla model, for reasons that are unexplained.  Given the weak results, I honestly feel that pause tokens are almost a distraction here.
>
> **(3)** Thank you for running the additional experiments in Appendix G, which is actually a very interesting result.  The authors write "training models from scratch is more sensitive to hyperparameter choices such as batch size".  However, that's not the conclusion that I would draw from this experiment.  By calculating the covariance matrix along sequence length, SeqVCR is ensuring that digits at _different positions in the sequence_ have diverse representations.  By using only the batch dimension, the SeqVCR loss improves diversity only among digits which occupy the _same position._  The fact that there is such a huge improvement in the first case says something interesting about the 5x5 multiplication problem.
>
> **(Conclusion)** To be honest, I have mixed feelings about this paper.  On the one hand, representation collapse is important, and SeqVCR in particular is an interesting technique.  On the other hand, I don't think the current paper really does it justice.  This has the potential to be a high-quality, high-impact paper if the authors properly explored the effect of SeqVCR on multiple tasks, including general language modeling and translation, did more ablations about the batch dimension vs. batch-dim + sequence-dim issue (Appendix G) on various tasks, etc.
>
> Instead, the authors restrict their experiments to extremely narrow downstream tasks, and then try to present pause tokens as an alternative to CoT, when pause tokens seem to have minimal impact on anything other than multi-digit multiplication.
>
> **Additional comments:**
>
> Figure 6 -- Should have an additional accuracy bar for SeqVCR + Pause.  I assume SeqVCR + Pause performs comparably to CoT because of Table 1, but that should also be shown in Figure 6.
>
> **Specific criticisms that have not yet been addressed (copied from before, with added emphasis).**
>
> Equation (3) defines the Seq-VCR loss. The text of the paper claims that it is "inspired by" prior work, and cites such work appropriately, but it is more than just "inspired". Equation (3) is lifted almost verbatim from the orginal VICReg (Bardes 2021) and VCReg (Zhu 2023) papers, and **the authors need to be crystal clear about the source of that equation in the text of the paper.**
>
> (As a minor nit, it is unclear to me whether or not the covariance term in equation (3) should have an additional 1/(d-1) factor; VICReg has the term, while VCReg does not. I would have appreciated it if the authors explained why they chose one version over the other.)
>
> For further clarity, **the authors should also devote a few lines (in the text of the paper) to defining how the covariance matrix C is computed**; as is done in other papers.  Otherwise, it can easily be confused with the cross-correlation matrix of the Barlow twins technique, which the authors also cite as inspiration.  _To further clarify -- I know you are using the covariance matrix, and not the cross-correlation matrix.  It might be helpful to other readers to have that spelled out by writing down the equation._

---

> > ### Author Response · Authors · 2024-12-01
> >
> > Dear Reviewer YSNL,
> >
> > As we approach the end of the discussion period, we’d like to follow up on our previous responses to ensure they fully addressed your comments. If you have any additional questions or concerns, we’re happy to provide further clarification.

---

### Official Review · Reviewer_DQGy · 2024-11-02

**Soundness:** 3
**Presentation:** 3
**Contribution:** 3
**Rating:** 6
**Confidence:** 3

**Summary:**

This paper focuses on the performance of decoder-only models on tasks such as multi-digit mathematical reasoning that require a series of immediate representations. They hypothesize representation collapse of intermediate layers as a key contributor to this poor performance, preventing effective storage of the intermediate steps necessary for these kinds of tasks. While chain of thought reasoning can be effective in counteracting this collapse and performing well on such tasks, the proposed approach seeks to increase entropy among intermediate layers and achieve similar performance with at a reduced computational cost. Formulated in terms of alpha-order matrix-based entropy, the formulate a regularization term which aims are increasing variance and decreasing covariance in the intermediate representations. Additionally, pause tokens are included in the method. Results on three kinds of tests are presented – computing arithmetic expressions, identifying the longest increasing integer subsequence, and performing multiplication of 4 digit or 5 digit numbers. Performance with the regularization term and pause tokens leads to performance which approaches chain of thought on most tests, and regularization performs well on its own for the simpler tasks.

**Strengths:**

The details of the method to seem to be heavily inspired by VICReg, but so far as I can judge, the application of it to the sequence/Transformer is original. The method is, in theory, computationally attractive compared to CoT and the results are fairly compelling.

The paper is clearly written and the quality of the presentation is moderately high.

**Weaknesses:**

The result in Table 1 naturally provokes a question: This and several previous studies show that GPT-2 with CoT performs remarkably well, but this is actually more difficult to achieve in larger models. What is the evidence/argument that the Seq-VCR approach will scale better with model size than CoT? Figure 8 hints at this but it doesn’t clearly address it.

The speedup vs CoT is intuitively reasonable but it would have been nice to see performance numbers as in the cite Deng 2024 paper.

Similarly, it would be helpful to understand the amount of hyperparameter optimization necessary for, e.g., identifying the number of pause tokens used to obtain the best results. Do the number of pause tokens necessary correlate with, e.g., task complexity?

For completeness, it would be nice to see CoT in figures 7 and 8.

**Questions:**

A discussion that clarified the improvement versus CoT would improve the significance, whether clearly establishing the speedup with Seq-VCR or showing its better generalization/scaling.

---

> ### Author Response · Authors · 2024-11-23
>
> Thank you for your detailed review and positive feedback, we address your concerns below.
>
> -**Scalability of Seq-VCR compared to CoT:**
> We appreciate this question. First, we would like to clarify that GPT3.5 and GPT4 results in Table 1 are from 5-shot prompted models, they are not fine-tuned models like GPT2. We believe that if GPT3.5 or GPT4 were fine-tuned with CoT they would achieve 100% results.
>
> Regarding increased model sizes, Figure 8 shows that Seq-VCR is effective on larger (deeper) models. For additional results on scaling experiments, please check the generic response (1).
>
> -**Inclusion of CoT in Figure 8:**
> Thank you for noticing that Figure 8 is missing the CoT numbers, we added them in the paper. They were 100% in all cases. Note, this is expected as CoT tokens carry more useful information than simple dummy <pause> tokens. However, <pause> combined with Seq-VCR serves as a competitive method while being computationally much cheaper and without using supervised CoT data for training.
>
> -**Speedup and Accuracy tradeoff for pause and CoT tokens:**
> We carried out this analysis in response to the reviewer’s feedback. Our method is computationally much faster compared to the CoT method. For further details, please refer to the computational complexity analysis in Generic Response (2).
>
> -**Hyperparameter optimization:**
> We conducted a manual search for hyperparameters, rather than performing exhaustive methods like grid search or random search. We found this approach sufficient to identify effective hyperparameters for our tasks. For the two coefficients, $\lambda_1$ and $\lambda_2$ in Eq. 3, we maintained their proportions close to those in [1]. Other hyperparameters, such as learning rate and batch size, are consistent with values used in related works [2, 3]. Further details are in Appendix A.
>
> -**# pause tokens vs. task complexity:**
> Based on the reviewer’s recommendation, we conducted additional experiments on arithmetic tasks using a fixed 5-layer model, varying the number of pause tokens (2, 4, 6, 8) and task complexity. We did not observe a clear correlation between task complexity and the number of pause tokens. We hypothesize that this may be due to all pause tokens sharing the same embedding. Future work will investigate the impact of using different embeddings for each pause token. These details are now included in Appendix B of the revised manuscript.
>
> [1] Bardes, Adrien, Jean Ponce, and Yann LeCun. "Vicreg: Variance-invariance-covariance regularization for self-supervised learning." arXiv preprint arXiv:2105.04906 (2021).
>
> [2] Deng, Yuntian, et al. "Implicit chain of thought reasoning via knowledge distillation." arXiv preprint arXiv:2311.01460 (2023).
>
> [3] Feng, Guhao, et al. "Towards revealing the mystery behind chain of thought: a theoretical perspective." Advances in Neural Information Processing Systems 36 (2024).
>
> We hope we addressed all of your concerns and would be very happy to answer any followup questions. We would like to request the reviewer to consider increasing the score.

---

> ### Public Comment · ~Victor_Prokhorov1 · 2024-11-25
> **Performance on the Arithmetic Dataset (Figure 4 and Figure 10)**
>
> Dear Authors,
>
> May I please ask you to clarify why the Vanilla model achieves a near perfect accuracy in Figure 4 (4 arithmetic operators) while slightly higher than 0.20 (zero pause tokens) in Figure 10 (again 4 arithmetic operators, the light blue colour)? Am I misreading it?

---

> > ### Author Response · Authors · 2024-11-25
> >
> > Thanks for pointing this out. We believe you are referring to Fig 7a) and Figure 10? We noticed that we did indeed add incorrect values for the vanilla model in Fig 10 in our appendix. We have updated this plot in our paper now and uploaded the revised version.

---

> > > ### Public Comment · ~Victor_Prokhorov1 · 2024-11-25
> > > **Figure 7 and Figure 10**
> > >
> > > Thank you very much for your prompt response. Yes, you are right Fig 7a. Make sense now. Also, for the Arithmetic Expressions dataset what number range did you use? This is a parameter (--number_range) that one uses to generate the dataset.

---

> > > ### Public Comment · ~Victor_Prokhorov1 · 2024-11-25
> > > **Figure 10**
> > >
> > > Maybe a small correction, in your analysis to Figure 10 you say: "with a **slight decrease** in accuracy as the number of pause tokens increases". If I am processing the figure correctly the performance of the Vanilla model (with 2 pause tokens and 4 operators) almost halves, would it be fair to say that adding the pause tokens to the Vanilla model significantly deteriorates its performance?  Do you have an intuition of why this is the case?

---

> > > > ### Author Response · Authors · 2024-11-29
> > > >
> > > > Apology for late response. We used 4 to 6 as the range for generating Arithmetic Expressions dataset.
> > > >
> > > > Thank you for pointing out the issue with the description of Figure 10. We have addressed this in the updated manuscript.
> > > >
> > > > In Figure 10, adding pause tokens significantly reduces the performance of the Vanilla model. We believe this occurs because the 5-layer model, even without pause tokens, is already capable of solving the 4-operator task. Adding pause tokens in such a scenario might distract the model in this relatively simple task. However, for more complex tasks like 5x5 digit multiplication, pause tokens enhance the model's performance while finetuning GPT2-Small.

---

> > > > > ### Public Comment · ~Victor_Prokhorov1 · 2024-12-02
> > > > > **Thanks!**
> > > > >
> > > > > Dear Authors,
> > > > >
> > > > > Thank you very much for your response and insights! Good luck with the submission!

---

> ### Author Response · Authors · 2024-12-01
>
> Dear Reviewer DQGy,
>
> As we approach the end of the discussion period, we’d like to follow up on our previous responses to ensure they fully addressed your comments. If you have any additional questions or concerns, we’re happy to provide further clarification.

---

> > ### Comment · Reviewer_DQGy · 2024-12-02
> >
> > I appreciate the substantial and thoughtful replies from the authors. I find myself agreeing with reviewer YSNL, who articulated better than me: its an important problem, its a very interesting approach, and yet I get stuck on whether the HPO and results really show that it will generalize beyond the cases considered.

---

### Official Review · Reviewer_STRW · 2024-11-03

**Soundness:** 3
**Presentation:** 3
**Contribution:** 3
**Rating:** 8
**Confidence:** 3

**Summary:**

This paper proposes a regularization technique for preventing representation collapse across the intermediate representations of a deep sequence model. Their results show that 1. the regularization technique increases matrix entropy (low matrix entropy = representation collapse) and 2. when pause tokens are added the language model significantly improved in performance for 4x4 and 5x5 arithmetic tasks.

**Strengths:**

- The paper presents a novel regularization technique that improves the model's performance in several reasoning tasks
- The paper presents detailed analysis of the experimental results, showcasing how exactly the regularization techniques affects the diversity of representations, the learning dynamics, as well as the digit-by-digit accuracy on multiplication tasks.

**Weaknesses:**

- The effect of the regularization technique was only studied for a relatively narrow domain of tasks, and it would be interesting to understand its effect on more general language benchmarks as well.
- Slightly more contextualization on how exactly pause tokens are incorporated would assist readers in understanding the work more easily as it is also a core part of what is being proposed in this work.

**Questions:**

Same as the weaknesses section.

---

> ### Author Response · Authors · 2024-11-23
>
> Thank you very much for your positive feedback on our work. We address your concerns in order below.
>
> -**The effect of the regularisation technique on the General Language Tasks:**
> This is a valid point raised by the reviewer. Additional Experiments and discussion are in Generic Response (1). Please check that.
>
> -**Contextualizition of pause tokens:**
> We really appreciate this feedback. We further discussed about pause token in the generic response (3) and also updating manuscript.
>
> We are happy to address if the reviewer has any further queries.

---

### Author Response · Authors · 2024-11-23
**General Responses 1/3 & 2/3**

We sincerely thank all reviewers for their time and valuable feedback on our work. In response, we conducted additional experiments using LLaMA 3.2-1B and Code-GPT2, which we are pleased to share.

****1) Representation Collapse in Large Language Models (LLMs)****

**New Experiments**: In response to your feedback, we fine-tuned LLaMA 3.2-1B on the 5x5 digit multiplication task, both with and without our proposed regularization.

**Results**: We evaluate representation collapse across layers in the fine-tuned model, both with and without our regularization. Our results show that the model trained with Seq-VCR exhibits a 15-40% increase in average entropy (indicating reduced collapse) compared to the baseline LLaMA model and solves the task by achieving an accuracy of 97.4%. Full details are provided in Figure 11 in Appendix I of the updated manuscript.

**Additional Scaling Analysis**: We also conducted more analysis on pre-trained LLaMA models of different sizes (1B, 3B, 8B) on the 5x5 digit multiplication dataset. We observed consistent representation collapse across all model sizes (Figure 12 in Appendix J). This indicates that the issue persists at different scales, emphasizing the importance of effective regularization techniques like Seq-VCR to reduce collapse and enhance intermediate reasoning capabilities.

****2) Generalizing Seq-VCR to a Broader Domain of Language Tasks****

**New experiments**: In response to your feedback, we conducted experiments on the CodeXGLUE text-to-code benchmark [1] using CodeGPT2, measuring representation collapse across layers in the fine-tuned model, both with and without our regularization.

**Results**: Our primary results show that the model trained with Seq-VCR exhibits a higher average entropy, indicating reduced representation collapse. Full details can be found in Figure 12 in Appendix J of the updated paper. Representation collapse in intermediate layers presents a significant challenge for Transformers in multi-step tasks, such as multiplication, where precise intermediate computations—like carries—are essential. This issue is also observed in the CodeXGLUE text-to-code benchmark, highlighting the potential of Seq-VCR to address these limitations.  Expanding its application to a wider range of NLP tasks, particularly those involving reasoning and generalization, presents a promising avenue for future research.

**Note:** We would also like to remind the reviewers that the scope we considered for this paper is for multi-step reasoning tasks and understanding the limit of current LLMs on why they fail in these tasks without explicit CoT supervision. While Seq-VCR may not show improvements in pure language tasks, such as translation, we believe it's still valuable to better understand LLMs from a representation point of view and for researchers concentrating on more reasoning-intensive problems. We also think future research along this line could be useful even for further increasing the representational capability of these models before using prompt-based techniques like CoT to fix it.

[1] https://github.com/microsoft/CodeXGLUE/tree/main/Text-Code/text-to-code

---

### Author Response · Authors · 2024-11-23
**General Response 3/3**

***3) Details on pause tokens***

Thank you for asking us to clarify the questions about pause tokens, we added the following clarifications, including the example in Section 3.5 of the updated paper in blue colored text, and added our speedup and accuracy analysis in Appendix E.

Increasing model capacity leads to a significant accuracy improvement in solving n × n digit multiplication tasks [1]. While some prior work increases depth [1] and chain-of-thought (CoT) [4] to enhance model capacity, an alternative approach is to use pause tokens[3], which act as explicit signals for the model to temporarily pause at intermediate states in sequential tasks before moving to the next computation step. We adopt pause tokens as a more cost-effective and computationally efficient alternative to Chain of Thought tokens, complementing our Seq-VCR approach. This combination enhances the representation capacity, allowing the model to better utilize the fixed-depth architecture. Below, we address the reviewers' queries regarding the placement, quantity, reason and time complexity considerations of pause tokens:

**Where:** In all experiments, pause tokens were placed between input and output tokens to emulate chain-of-thought (CoT) reasoning. This setup allowed the model to take "pauses," improving its ability to organize intermediate representations and enhance computational reasoning. For example, the input-output format could look like <question> </pause_start> <pause> <pause> </pause_end> <answer>.

**How many:** We tried 2, 4, 6, 8  pause tokens on 4x4 and 5x5 both digit multiplication tasks, and arithmetic tasks and we did not find any correlation with task complexity. This result is shown in Fig 10 in the Appendix C. We believe it may be due to the fact that all the pause tokens share the same embedding. Future work will explore the effect of having different embeddings per pause tokens.

**Under what circumstances:** When CoT instructions are unavailable or when inference time needs to be reduced, pause tokens provide a simple and effective solution. Unlike CoT, which requires extensive labeled data for multi-step reasoning, Seq-VCR uses a few dummy pause tokens to solve tasks like multiplication in a fraction of the time, while performing at a similar accuracy close to 100% in our experiments (see table below).
Seq-VCR's efficiency in both inference time, data requirements, and accuracy makes it a scalable and robust approach compared to CoT.

**Computation Complexity Analysis:** We provide a detailed analysis below, which we also added in Appendix F of the revised document. Seq-VCR offers notable benefits over Chain-of-Thought (CoT) reasoning, particularly in reducing inference time and dependency on costly human-supervised data. Unlike CoT, which requires extensive labeled data for multi-step reasoning, Seq-VCR uses a few dummy pause tokens to solve tasks like multiplication in a fraction of the time, while performing at a similar accuracy close to 100% in our experiments (see table below).

To compute inference time we utilize the normalized throughput, using the following equation, as introduced the deng2024explicit[2] paper:
$$ T_{\text{norm}} = \frac{T_{\text{target}}}{T_{\text{base}}} $$
Here:
- $$T_{\text{norm}}  \text{is the normalized throughput, which represents the relative inference speed.}$$
- $$T_{\text{target}}  \text{is the throughput (number of examples processed per second) when using target method.}$$
- $$T_{\text{base}}  \text{is the throughput (number of examples processed per second) for the baseline model without Chain of Thought or Pause tokens.}$$

Normalized Throughput (the higher, the better) on 4x4 and 5x5 digit multiplication without CoT tokens, with CoT tokens, and with 2 pause tokens.

|  **Method**   | T_{norm}  |   | Accuracy |  |
|------------------------|---------|---------|----------|----------|
|  **Method**   |  (*4x4*) |  (*5x5*)  | (*4x4*)  | (**5x5**)   |
|------------------------|---------|---------|----------|----------|
| No CoT                 | 1.0     | 1.0     | 0.25     | 0.0      |
| With CoT               | 0.17    | 0.14    | 1.0      | 1.0      |
| Seq-VCR + Pause (2)    | 0.95    | 0.91    | 0.992    | 0.995    |


[1] Qiu, Luyu, et al. "Dissecting Multiplication in Transformers: Insights into LLMs." arXiv preprint arXiv:2407.15360 (2024).

[2] Deng, Yuntian, et al. "Implicit chain of thought reasoning via knowledge distillation." arXiv preprint arXiv:2311.01460 (2023).

[3] Goyal, Sachin, et al. "Think before you speak: Training language models with pause tokens." arXiv preprint arXiv:2310.02226 (2023).

[4] Wei, Jason, et al. "Chain-of-thought prompting elicits reasoning in large language models." Advances in neural information processing systems 35 (2022): 24824-24837.

---

### Meta-Review · Area_Chair_xayY · 2024-12-23

**Metareview:**

This paper proposes a regularization technique for preventing representation collapse across the intermediate representations of a deep sequence model. Their results show that 1. the regularization technique increases matrix entropy (low matrix entropy = representation collapse) and 2. when pause tokens are added the language model significantly improved in performance for 4x4 and 5x5 arithmetic tasks.

The strengths of this paper are:
* simple technique
* effectiveness on some simple reasoning tasks the authors experimented on

Weakness of this paper:
* unsure about how this technique would generate to leading LLMs (though the author added experiments to LLama during rebuttal)
* unsure about how this technique performs on more complex reasoning tasks.
* This method requires training, hence should be compared to any other reasoning augmentation method that also requires training.

**Additional Comments On Reviewer Discussion:**

The authors did a good rebuttal to address most of the reviewers' questions.

---

### Decision · Program_Chairs · 2025-01-22

Accept (Poster)